# Nitrogen oxides in the global upper troposphere: interpreting cloud-sliced $NO_2$ observations from the OMI satellite instrument

Eloise A. Marais[1,2,*], Daniel J. Jacob[2,3], Sungyeon Choi[4], Joanna Joiner[4,5], Maria Belmonte-Rivas[6], Ronald C. Cohen[7,8], Steffen Beirle[9], Lee T. Murray[10], Luke D. Schiferl[11,**], Viral Shah[12], Lyatt Jaeglé[12]

[1]School of Geography, Earth, and Environmental Sciences, University of Birmingham, Birmingham, UK.
[2]John A Paulson School of Engineering and Applied Sciences, Harvard University, Cambridge, MA, USA.
[3]Earth and Planetary Sciences, Harvard University, Cambridge, MA, USA.
[4]Science Systems and Applications Inc., Lanham, MD.
[5]NASA Goddard Space Flight Center, Greenbelt, MD.
[6]Royal Netherlands Meteorology Institute, De Bilt, the Netherlands.
[7]Department of Chemistry, University of California at Berkeley, Berkeley, CA.
[8]Department of Earth and Planetary Science, University of California at Berkeley, Berkeley, CA.
[9]Max-Planck-Institut für Chemie, Mainz, Germany.
[10]Department of Earth and Environmental Sciences, University of Rochester, Rochester, New York, USA.
[11]Department of Civil and Environmental Engineering, Massachusetts Institute of Technology, Cambridge, USA.
[12]Department of Atmospheric Sciences, University of Washington, Seattle, WA, USA.
* Now at: Department of Physics and Astronomy, University of Leicester, Leicester, UK.
** Now at: Lamont-Doherty Earth Observatory, Columbia University, Palisades, NY, USA.

*Correspondence to*: Eloise A. Marais (eloise.marais@le.ac.uk)

**Abstract.** Nitrogen oxides ($NO_x \equiv NO + NO_2$) in the upper troposphere (UT) have a large impact on global tropospheric ozone and OH (the main atmospheric oxidant). New cloud-sliced observations of UT $NO_2$ at 450-280 hPa (~6-9 km) from the OMI satellite instrument produced by NASA and KNMI provide global coverage to test our understanding of the factors controlling UT $NO_x$. We find that these products offer useful information when averaged over coarse scales ($20° \times 32°$, seasonal), and that the NASA product is more consistent with aircraft observations of UT $NO_2$. Correlation with LIS/OTD satellite observations of lightning flash frequencies suggests that lightning is the dominant source of $NO_x$ to the upper troposphere except for extratropical latitudes in winter. The $NO_2$ background in the absence of lightning is 10-20 pptv. We infer a global mean $NO_x$ yield of $280 \pm 80$ moles per lightning flash, with no significant difference between the tropics and mid-latitudes, and a global lightning $NO_x$ source of $5.9 \pm 1.7$ Tg N $a^{-1}$. There is indication that the $NO_x$ yield per flash increases with lightning flash footprint and with flash energy.

## 1. Introduction

Nitrogen oxides ($NO_x \equiv NO + NO_2$) in the upper troposphere (UT) have profound effects on the oxidizing capacity of the atmosphere and on climate, but the factors controlling their concentrations are poorly understood. $NO_x$ in the UT impacts climate by efficiently producing ozone where it is a potent greenhouse gas (Dahlmann et al., 2011; Worden et al., 2011; Rap et al., 2015) and by increasing the concentration of OH (the main tropospheric oxidant) (Murray et al., 2012; Murray et al., 2014). Primary $NO_x$ sources in the UT include lightning, aircraft, convective injection, and downwelling from the stratosphere (Ehhalt et al., 1992; Jaeglé et al., 1998; Bertram et al., 2007). $NO_x$ cycles chemically with reservoir species including nitric acid ($HNO_3$), pernitric acid ($HNO_4$), dinitrogen pentoxide ($N_2O_5$), peroxyacetylnitrate (PAN), and other organic nitrates, thus defining the $NO_y$ chemical family ($NO_y \equiv NO_x$ + reservoirs). Effective loss of $NO_x$ from the UT is through subsidence of $NO_y$ to lower altitudes where deposition of $HNO_3$ provides the ultimate sink. The residence time of $NO_y$ in the UT is 10-20 days (Prather and Jacob, 1997). The lifetime of

NO$_x$ against conversion to short-lived reservoirs varies from ~3 hours in the convective outflow of thunderstorms to 0.5-1.5 days

in background air (Nault et al., 2016). Chemical recycling from these reservoirs maintains relatively high UT NO$_x$ background concentrations (Bradshaw et al., 2000; Baehr et al., 2003; Nault et al., 2016).

Representation of lightning NO$_x$ in chemical transport models (CTMs) is particularly uncertain. Physically-based parameterizations relating lightning frequency to deep convective cloud tops, convective mass flux, convective precipitation, or high-cloud ice

content have poor predictive capability (Tost et al., 2007; Allen et al., 2010; Murray et al., 2012; Finney et al., 2014), limiting our ability to estimate the response of lightning NO$_x$ to future climate (Finney et al., 2016; 2018). An alternative is to prescribe flash densities from space-based observations and static NO$_x$ production rates per flash (Sauvage et al., 2007; Allen et al., 2010; Murray et al., 2012). NO$_x$ production efficiencies per flash in the literature vary from <10 to 5000 moles nitrogen per flash (mol N fl$^{-1}$) (Schumann and Huntrieser, 2007; Murray, 2016). Global chemical transport models (CTMs) typically use 100-500 mol N fl$^{-1}$,

sometimes assuming higher production rates at mid-latitudes than in the tropics (Hudman et al., 2007; Ott et al., 2010), and a global lightning NO$_x$ source of 3-7 Tg N a$^{-1}$ to match observations of tropospheric ozone and NO$_y$ species (Sauvage et al., 2007).

Our understanding of UT NO$_x$ has so far been evaluated with observations from aircraft campaigns (Drummond et al., 1988; Jacob et al., 1996; Crawford et al., 1997; Jaeglé et al., 1998; Bradshaw et al., 2000; Hudman et al., 2007; Stratmann et al., 2016). There

are also long-term NO$_x$ measurements from instruments onboard commercial aircraft dating back to the 1990s, but these are mostly over the north Atlantic and the NO$_2$ measurements have low precision and interference from thermally unstable NO$_x$ reservoir compounds (Brunner et al., 2001). A number of studies have used satellite observations of tropospheric NO$_2$ columns from solar backscatter to infer lightning NO$_x$ emissions (Beirle et al., 2010; Pickering et al., 2016; Bucsela et al., 2010), including in combination with global models (Boersma et al., 2005; Martin et al., 2007; Miyazaki et al., 2014). These studies estimate global

lightning NO$_x$ emission of 1 to 8 Tg N a$^{-1}$.

New cloud-sliced satellite products of tropospheric NO$_2$ mixing ratios at 280-450 hPa (~6-9 km) offer additional vertical resolution by retrieving partial NO$_2$ columns above clouds and exploiting differences in heights of neighboring clouds to calculate NO$_2$ mixing ratios (Choi et al., 2014; Belmonte-Rivas et al., 2015). There are two new products of seasonal mean UT NO$_2$ mixing ratios

retrieved from Ozone Monitoring Instrument (OMI) partial NO$_2$ columns by research groups at KNMI and NASA. The KNMI product has been evaluated against UT NO$_2$ from the Tracer Model version 4 (TM4) CTM. Large regional differences between OMI and TM4 are attributed to model deficiencies in lightning NO$_x$ and uplift of anthropogenic pollution (Belmonte-Rivas et al., 2015). The NASA UT product is new to this work and follows a similar retrieval approach to the mid-tropospheric (900-650 hPa) product of Choi et al. (2014). That product was evaluated with aircraft observations of NO$_2$ and interpreted with the Global

Modeling Initiative (GMI) CTM (Choi et al., 2014). Choi et al. (2014) identified large discrepancies between modeled and observed NO$_2$ seasonality over regions influenced by pollution and lightning.

Here we compare the two UT NO$_2$ products, obtained with distinct retrieval methods, and use aircraft observations of NO$_2$ from multiple NASA DC8 aircraft campaigns to arbitrate and evaluate the information that can be derived from the satellite datasets.

We go on to test current understanding of UT NO$_x$ and the implications for lightning emissions using the GEOS-Chem CTM.

## 2. OMI observations of upper troposphere NO$_2$

OMI is onboard the NASA Aura satellite launched into sun-synchronous orbit in July 2004. It has an overpass time of about 13h30 local time (LT), a swath width of 2600 km, and a horizontal resolution of 13 km × 24 km at nadir (Levelt et al., 2006). Columns of $NO_2$ along the instrument viewing path (slant columns) are obtained by spectral fitting of solar backscattered radiation in the 405-465 nm window (Boersma et al., 2011). Standard products of total and tropospheric column $NO_2$ are screened for cloudy scenes using a cloud radiance fraction threshold of 0.5. Partial columns of $NO_2$ above cloudy scenes can be used to estimate vertically resolved $NO_2$ mixing ratios, as was first demonstrated with satellite observations of ozone (Ziemke et al., 2001). This approach, so-called cloud slicing, assumes a uniform trace gas concentration between two horizontally nearby clouds at different altitudes, so that the UT $NO_2$ mixing ratio is proportional to the slope of the partial columns versus the corresponding cloud pressures at the optical centre of the cloud. Two products of seasonal mean UT $NO_2$ have been retrieved from OMI following distinct retrieval steps detailed below: a product from KNMI at 330-450 hPa for 2006 (Belmonte-Rivas et al., 2015) and from NASA at 280-450 hPa for 2005-2007 following an approach similar to that used to retrieve mid-tropospheric $NO_2$ (Choi et al., 2014). In what follows we distinguish the two OMI $NO_2$ products as KNMI and NASA.

The KNMI product uses DOMINO v2.0 slant columns (Boersma et al., 2011) and OMCLDO2 cloud fractions and altitudes (Acarreta et al., 2004) over partially to very cloudy scenes (cloud radiance fraction > 0.5). Contamination due to $NO_2$ from below (up to 66% over polluted land masses) is estimated using the TM4 model and removed. Stratospheric $NO_2$ from an assimilated product (Belmonte-Rivas et al., 2014) is also removed. An air mass factor (AMF) (detailed in Boersma et al. (2004)) that accounts for viewing geometry, surface albedo, light attenuation by gases along the viewing path, and sensitivity to $NO_2$ vertical distribution is applied to the resultant partial slant columns to convert to vertical columns. Additional data filtering removes scenes with solar zenith angle (SZA) $\geq 70°$ and surface albedo $\geq 30\%$. Resultant daily vertical partial columns are aggregated on consistent pressure and horizontal ($1° \times 1°$) grids and used to determine seasonal mean UT $NO_2$ mixing ratios for gridsquares with at least 30 measurements. UT $NO_2$ centred at 380 hPa (range 330-450 hPa) is estimated as the difference between partial tropospheric columns retrieved above two neighboring clouds with cloud pressures in the ranges 330-450 hPa and 380-500 hPa, respectively. Biases from sampling cloudy scenes, such as the effect of clouds on photochemistry, are corrected using TM4. These are small (typically <20%) in the UT (Belmonte-Rivas et al., 2015).

The NASA UT $NO_2$ product for 2005-2007, centred at 350 hPa (~280-450 hPa), uses updated version 3 slant columns (OMNO2 v3.0) (Krotkov et al., 2017) that correct for a positive bias in the DOMINO v2.0 product with improved spectral fitting (Marchenko et al., 2015; van Geffen et al., 2015). Partial columns from the cloud height to the top of the atmosphere are retrieved for individual OMI pixels above very cloudy scenes (cloud radiance fraction > 0.7) to minimize contamination from below. Cloud fraction and height is from the OMCLDO2 product (Acarreta et al., 2004). The AMF accounts for viewing path geometry and light scattering by clouds with uniform scatter that are optically thick and geometrically thin (near-Lambertian clouds). Data filtering is applied to remove scenes with SZA > 80°, snow/ice cover, and severe aerosol pollution that could be misclassified as clouds. Daily UT $NO_2$ is estimated for neighboring partial columns with sufficient cloud variability (cloud pressure distance > 160 hPa) and well-mixed $NO_2$ ($NO_2$ vertical gradient < 0.33 pptv hPa$^{-1}$ diagnosed with the GMI CTM). The stratospheric column is assumed uniform above neighboring clouds and so is removed when differencing two nearby partial columns. Daily values of UT $NO_2$ are gridded to obtain seasonal means at $5° \times 8°$ (latitude × longitude) for scenes with at least 50 measurements. Gaussian weighting is applied to assign higher weighting to UT $NO_2$ closest to 350 hPa. Choi et al. (2014) used a similar approach to retrieve mid-tropospheric $NO_2$ except that cloud fraction and height were from the OMCLDRR product, and successful retrieval required a stricter cloud radiance fraction of 0.9, a minimum of 30 measurements, and a wider minimum cloud pressure distance of 200 hPa. A shift in cloud radiance fraction

threshold from 0.9 (Choi et al., 2014) to 0.7 (this work) only introduces a small (<5%) difference in the retrieved partial columns due to contamination from below, as estimated by Pickering et al. (2016) for OMI scenes over the Gulf of Mexico.

120    Figure 1 compares seasonal mean UT $NO_2$ from the two satellite products in December-February and June-August. KNMI $NO_2$ is gridded to the NASA coarse grid. Data for March-May and September-November are in the Supplement (Figure S1). KNMI $NO_2$ has greater coverage than the NASA product, due to a lower cloud fraction threshold in the retrieval. The two products exhibit very different spatial features. Spatial correlation between the two products (Pearson's correlation coefficient between coincident gridsquares) is R = 0.41 in December-February and R = 0.38 in June-August. There is marginal improvement in the correlation

125    with further spatial averaging. At 20° × 32° we find R = 0.50 in December-February and R = 0.45 in June-August. The correlation only increases substantially in September-November from R = 0.49 at 5° × 8° (Figure S1) to R = 0.66 at 20° × 32°. KNMI is systematically lower than NASA in all seasons for coincident gridsquares, varying from 16% lower in June-August to 48% lower in December-February at 20° × 32°.

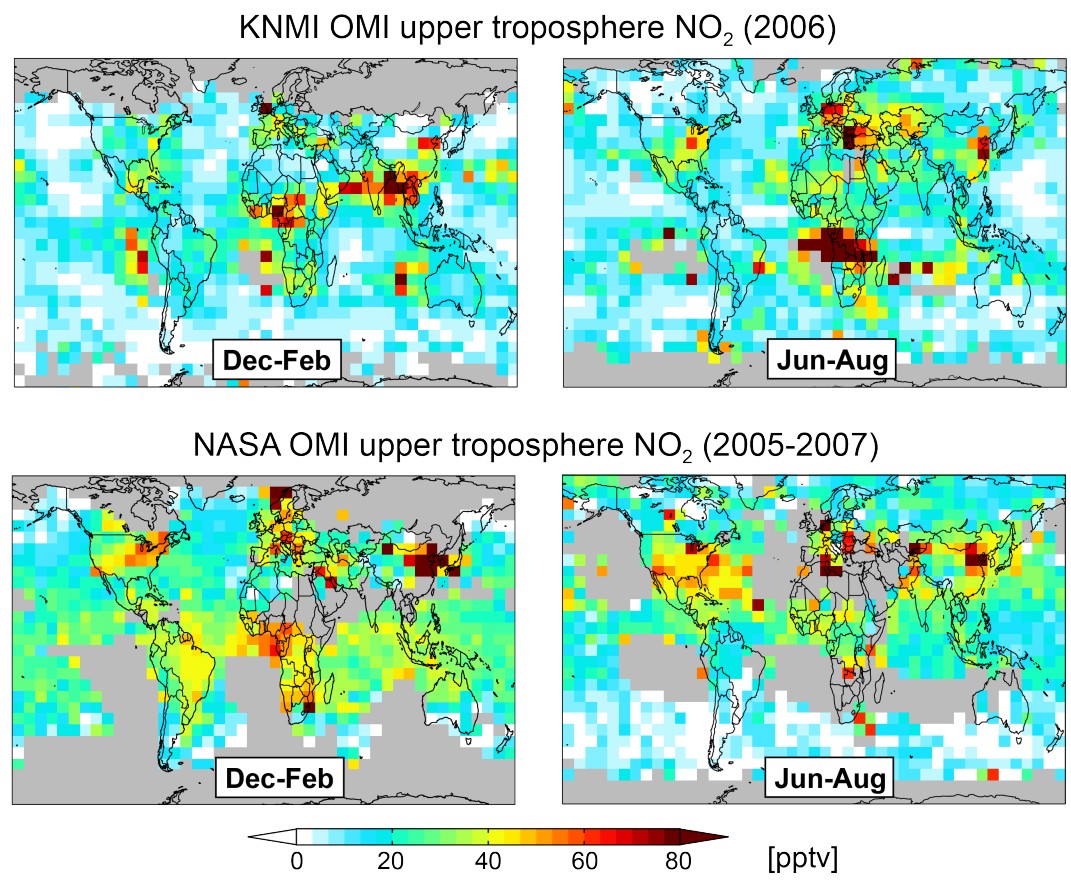

**Figure 1. Upper troposphere (UT) $NO_2$ from the OMI satellite instrument. Seasonal mean UT $NO_2$ from KNMI in 2006 at 330-450 hPa (top) is compared to NASA in 2005-2007 at 280-450 hPa (bottom). Data are at 5° × 8° horizontal resolution for December-February (left) and June-August (right). Grey areas indicate no data and, for NASA, scenes with fewer than 50 measurements.**

135    Contamination of UT $NO_2$ from below the cloud may still be present in the datasets despite attempts to correct for this using the TM4 model in the case of KNMI and by only considering very cloudy scenes in the case of NASA. These include a large enhancement in KNMI $NO_2$ (> 90 pptv) over southern Africa in June-August when there is intense biomass burning, and the $NO_2$ hotspot over northeast China in all seasons in both products (Figures 1, S1). Belmonte-Rivas et al. (2015) caution that the contamination correction in the KNMI product relies on accurate simulation of $NO_2$ vertical distribution.

## 3. Evaluation of OMI upper troposphere NO₂ with aircraft observations

We evaluate the OMI UT NO₂ products with observations from NASA DC8 aircraft campaigns over North America and Greenland in spring-summer, for which dense coverage is available (Figure 2). These include the INTEX-A, INTEX-B, ARCTAS, DC3, and SEAC⁴RS campaigns. Only INTEX-B is in the same year as the OMI products but we consider interannual variability to be only a small source of error. All NO₂ measurements are from thermal-dissociation laser-induced fluorescence (TD-LIF) instruments (Day et al., 2002). These are susceptible to interference from decomposition of thermally unstable reservoir compounds including methyl peroxy nitrate ($CH_3O_2NO_2$) and $HNO_4$ (Browne et al., 2011). Publicly available DC3 and SEAC⁴RS TD-LIF NO₂ are already corrected for this interference. We apply a correction for the other campaigns using the relationship between temperature and percentage interference from Browne et al. (2011). Observed mean ambient air temperature in the UT during INTEX-A was 246 K, corresponding to 20% interference. That for INTEX-B was 241 K (30% interference) and 236 K for ARCTAS (38% interference).

There are also NO₂ observations from the recent NASA ATom campaign, and from the In-service Aircraft for a Global Observing System (IAGOS) commercial aircraft campaign (Berkes et al., 2017). These use chemiluminescence instruments that are also susceptible to interference. Chemiluminescence and TD-LIF NO₂ are consistent during the SEAC⁴RS campaign for the altitude range considered in this work (6-9 km) (Travis et al., 2016), but the interference from chemiluminescence is challenging to quantify, due to dependence on the operator and instrument design that varies across campaigns (Reed et al., 2016).

Figure 2 shows the sampling extent of TD-LIF UT NO₂ over North America and Greenland in spring-summer at 450-280 hPa around the satellite overpass (11h00-16h00 LT) for scenes not influenced by the stratosphere (diagnosed with collocated ozone/CO > 1.25 mol mol$^{-1}$ (Hudman et al., 2007)). Concentrations of UT NO₂ exceed 80 pptv over the eastern US due to lightning NOₓ emissions and convective transport of boundary layer pollution, and are < 30 pptv over the rest of the domain.

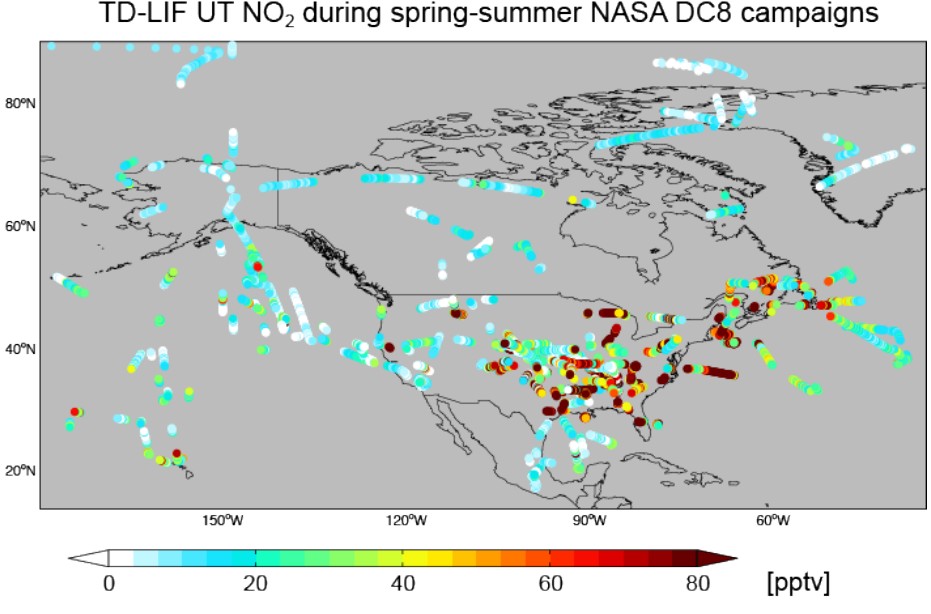

**Figure 2. NASA DC8 upper troposphere NO₂ over North America in spring-summer (March-August). Observations are from the TD-LIF instrument at 450-280 hPa, 11h00-16h00 local time, and without stratospheric influence. Campaigns include INTEX-A in June-August 2004 (Singh et al., 2006), INTEX-B in March-May 2006 (Singh et al., 2009), ARCTAS in March-April and June-July 2008 (Jacob et al., 2010), DC3 in May-June 2012 (Barth et al., 2015), and SEAC⁴RS in August 2013 (Toon et al., 2016).**

Figure 3 shows the spatial correlation between the March-August mean gridded aircraft data and the OMI UT NO$_2$ from the KNMI and NASA products as a function of horizontal resolution. There is no significant spatial correlation between the OMI products and aircraft NO$_2$ at 5° × 8° (R < 0.1) and 10° × 16° (R < 0.2). The correlation improves with further spatial averaging, peaking at 20° × 32° (R = 0.56 for KNMI, R = 0.64 for NASA). The satellite products are also spatially consistent at this resolution (R = 0.89), but KNMI is 43% lower than NASA.

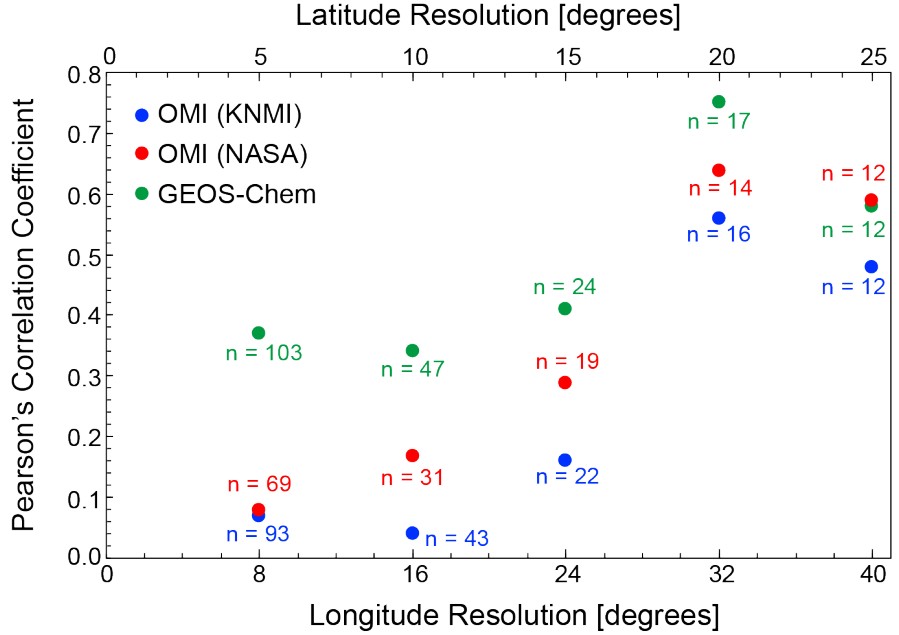

**Figure 3. Evaluation of OMI and GEOS-Chem upper troposphere NO$_2$ with aircraft observations. Individual points are Pearson's correlation coefficients between gridded March-August mean UT NO$_2$ measured from the aircraft and OMI KNMI in 2006 (blue), OMI NASA in 2005-2007 (red), and GEOS-Chem in 2006 (green), for grid averaging domains of 5° × 8° (latitude × longitude), 10° × 16°, 15° × 24°, 20° × 32°, and 25° × 40°. Values inset are the number of points at each resolution. The domain sampled is shown in Figure 2.**

Figure 4 compares the spatial distrbution of OMI and aircraft UT NO$_2$ at 20° × 32° over North America. Domain mean KNMI UT NO$_2$ is 38% lower than the aircraft observations, compared to 2.2% higher for NASA UT NO$_2$. Both products exhibit less variability (reduced major axis, RMA, regression slopes < 1) and high bias in background NO$_2$ compared to the aircraft observations (positive RMA intercepts of 5.9 ± 1.4 pptv for KNMI and 9.2 ± 2.7 pptv for NASA). We proceed with the NASA UT NO$_2$ product at 20° × 32°, as correlation peaks at this resolution and the NASA product is more consistent with domain mean aircraft UT NO$_2$ than the KNMI product.

## 4.  Constraints on upper tropospheric NO$_x$

The NASA product provides near-global coverage of UT NO$_2$ to assess current understanding of regional UT NO$_x$ sources and dynamics by comparing to UT NO$_2$ from the GEOS-Chem CTM (version 10-01; http://wiki.seas.harvard.edu/geos-chem/index.php/GEOS-Chem_v10-01) driven with NASA MERRA-2 reanalysis meteorology. The model horizontal resolution is 2° × 2.5° and the output is regridded to 20° × 32° for comparison with OMI. GEOS-Chem is sampled under all-sky conditions in the satellite overpass window (12h00-15h00 local time). We find that the effect on NO$_2$ of sampling the model under cloudy conditions is small. Isolating NO$_2$ under very cloudy conditions using MERRA-2 cloud fractions decreases modeled UT NO$_2$ by no more than 5 pptv in the tropics/subtropics and less at higher latitudes. We use output from the model for 2006 following a one-

year spin-up for chemical initialization. Interannual variability in UT $NO_2$, determined as the difference between modeled 2006 and multi-year mean (2005-2007) UT $NO_2$, is small (< 4 pptv) everywhere except central Africa year-round (4-12 pptv), the Arctic north of 60°N (up to 25 pptv), and the Middle East in June-August and northern India in March-May (both 10-20 pptv). Recent evaluation of model $NO_2$ with observed vertical profiles from the SEAC[4]RS aircraft campaign show no significant bias in the 6-9 km range of the OMI product (Travis et al., 2016; Silvern et al., 2018).


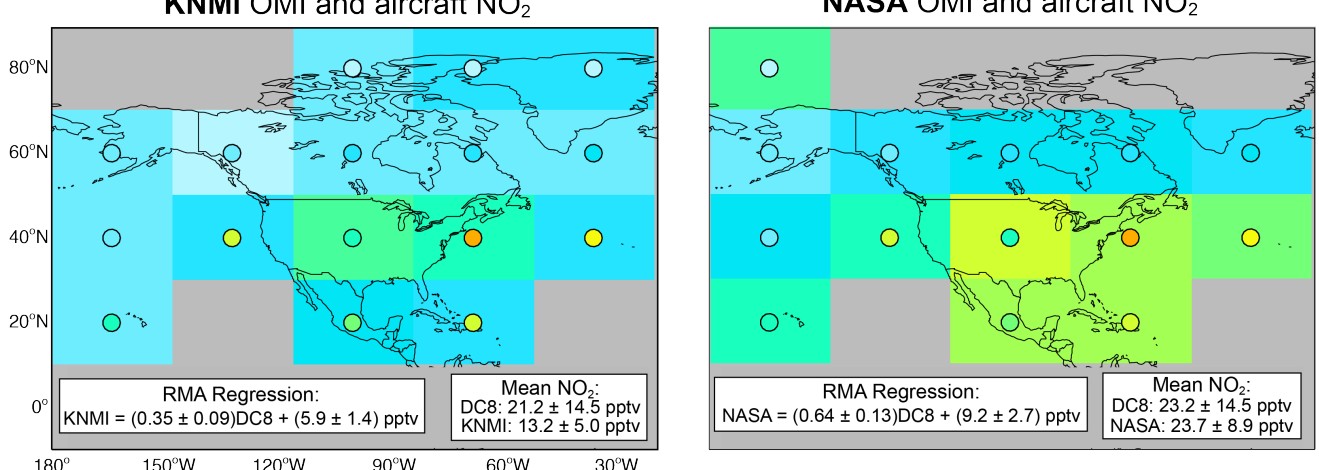

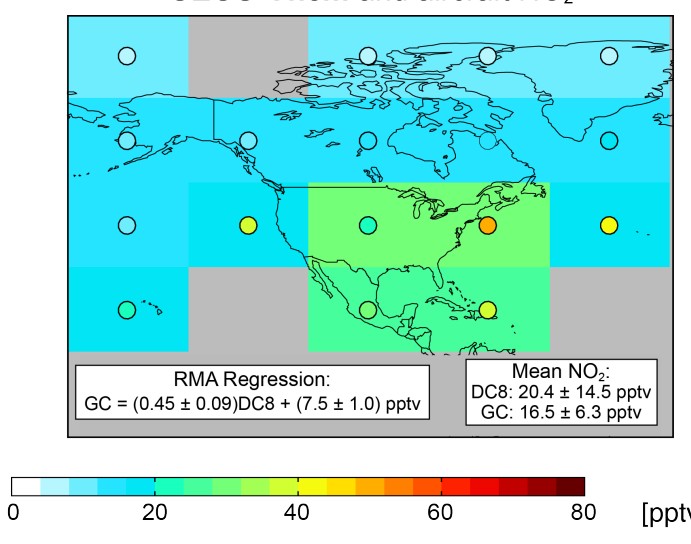

**Figure 4. March-August upper troposphere $NO_2$ over North America. All data are at 20° × 32°. Background colors in the different panels show concentrations from KNMI, NASA, and GEOS-Chem (GC). Circles show the aircraft observations (same in all panels). Aircraft observations are for 11h00-16h00 local time (LT). The model is sampled in the satellite**
**overpass time window (12h00-15h00 LT). Model and aircraft data are at 280-450 hPa and screened for stratospheric influence using ozone/CO > 1.25 mol mol[-1]. Inset boxes show reduced major axis (RMA) regression statistics and mean $NO_2$ for coincident gridsquares. Grey gridsquares indicate no observations.**

Local GEOS-Chem emissions of $NO_x$ in the UT include aircraft and lightning. Aircraft emissions from the AEIC inventory (Stettler et al., 2011) total 0.82 Tg N in 2006; much less than lightning in the same year (6.5 Tg N). Transport from the stratosphere is
simulated using a climatology of $NO_y$ species concentrations from the GMI model above the tropopause (Murray et al., 2012) and is very small (0.4 Tg N a[-1] as total $NO_y$). Lightning in the model is estimated using the parameterization implemented by Murray et al. (2012). This includes an initial estimate of lightning flashes using the Price and Rind (1992, 1993, 1994) relationship between cloud-top height and lightning flashes. These are then scaled to the same annual global flash frequency (46 fl s[-1]) and regional

distribution as the climatology from the combined Lightning Imaging Sensor (LIS) and Optical Transient Detector (OTD) high-

resolution monthly climatology (LIS/OTD HRMC) (Cecil et al., 2014). The standard GEOS-Chem model has higher $NO_x$ yields per flash at northern mid-latitudes (north of 35°N) than in the tropics (500 mol N $fl^{-1}$ versus 260 mol N $fl^{-1}$), but we find that this overestimates observed OMI UT $NO_2$ by 10-20 pptv (20-40%) at northern mid-latitudes in summer when the lightning source is dominant. Here we address this overestimate by assuming a $NO_x$ yield of 260 mol N $fl^{-1}$ everywhere. This decreases global lightning $NO_x$ emissions by 15% from 6.5 to 5.5 Tg N $a^{-1}$. The lightning parameterization in GEOS-Chem does not distinguish lightning

$NO_x$ production from flashes within or between clouds (intra- or inter-cloud) or from the cloud to the Earth's surface (cloud-to-ground).

Figure 3 shows the spatial correlation between the model and aircraft observations. The model is more consistent with the aircraft observations than OMI at fine spatial resolution. Like OMI, GEOS-Chem correlation with the aircraft observations improves with

spatial averaging, peaking at 20° × 32° (R = 0.75). Figure 4 also shows comparison of March-August GEOS-Chem UT $NO_2$ with the aircraft observations at 20° × 32°. The model is sampled over the same pressure range as NASA (280-450 hPa) around the OMI overpass (12h00-15h00 LT) and is filtered for stratospheric influence using model ozone/CO > 1.25 mol $mol^{-1}$. Domain average UT $NO_2$ from the model is 19% lower than the aircraft measurements and the model also overestimates background UT $NO_2$ (intercept = 7.5 ± 1.0 pptv) and underestimates the variability (slope = 0.45 ± 0.09).


Figure 5 compares seasonal mean OMI and GEOS-Chem UT $NO_2$ in December-February and June-August. The other seasons are shown in the Supplement (Figure S2). Formation of PAN, $HNO_4$ and $CH_3O_2NO_2$ accounts for over 75% of $NO_x$ loss in the model in all seasons. Lower concentrations of UT $NO_2$ in the northern hemisphere winter compared to summer in the model is mostly because lightning activity is at a minimum. The model underestimates UT $NO_2$ in the northern mid-latitudes in winter by 20-40

pptv, suggesting misrepresentation of another process in the model, such as excessive $NO_x$ loss by $N_2O_5$ hydrolysis in aerosols (Kenagy et al., 2018). The particularly large bias over polluted regions in winter could also be due to contamination of the UT $NO_2$ retrievals by enhanced boundary layer $NO_2$.

### NASA OMI seasonal mean upper troposphere NO₂

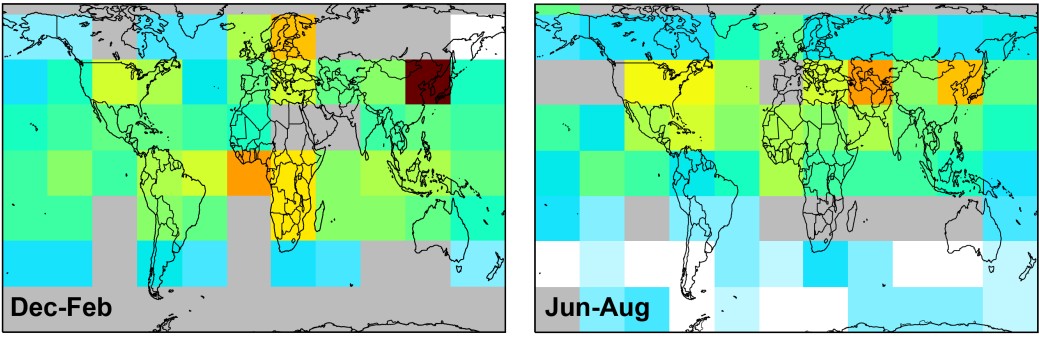

### GEOS-Chem seasonal mean upper troposphere NO₂

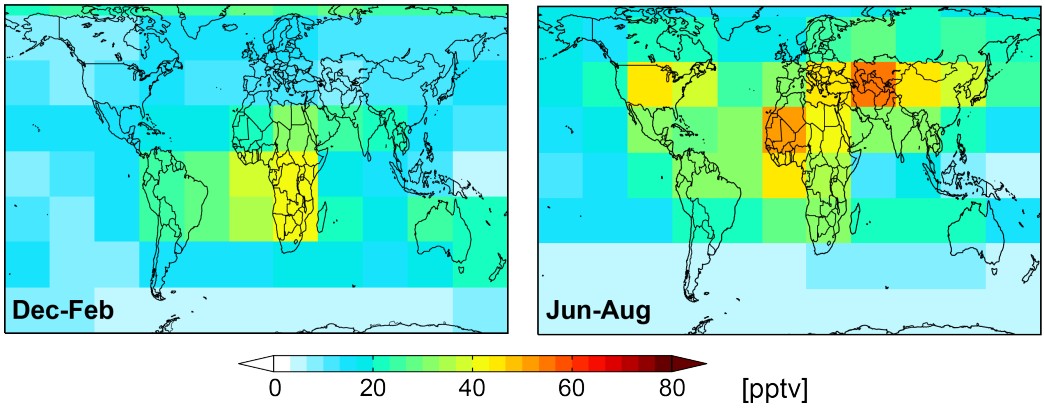

**Figure 5. Observed and modelled upper troposphere NO₂.** The figure shows NASA OMI seasonal mean UT NO₂ for 2005-2007 (top) and corresponding GEOS-Chem model values (bottom). The model is sampled at 280-450 hPa during the satellite overpass (12h00-15h00 LT), and filtered for stratospheric influence. Data are at 20° × 32° horizontal resolution for December-February (left) and June-August (right). Grey gridsquares in the top panel indicate no OMI data.

Figure 6 shows the log-log relationship between seasonal mean LIS/OTD lightning flash climatology and seasonal mean UT NO₂ from OMI and GEOS-Chem, and the corresponding reduced major axis linear regression fits. Data are divided into northern mid-latitudes and tropics. We exclude the contaminated observations over northeast China and the wintertime northern mid-latitude gridsquares that show no correlation with lightning flashes (R < 0.1). Results from multi-model sensitivity studies indicate that UT NO$_x$ in winter is predominantly from surface sources, with a smaller contribution from extra-tropical lightning (Grewe et al., 2001). Background concentration of UT NO₂ (intercepts in Figure 6) from non-lightning sources is 10-20 pptv and is 3-5 pptv higher in the northern mid-latitudes than the tropics. The slopes for the linear fits to lightning flash frequency are consistent between the OMI observations and GEOS-Chem, and show similar slopes for northern mid-latitudes and the tropics. Fitting the ratio between OMI observations and GEOS-Chem on the 20° × 32° grid implies a NO$_x$ yield per flash of 280 ± 80 mol N fl$^{-1}$ with no significant difference between mid-latitudes and the tropics, and no significant difference with the GEOS-Chem prior estimate of 260 mol N fl$^{-1}$. Out prior estimate of global lightning source was 5.5 Tg N a$^{-1}$, and the improved estimate is 5.9 ± 1.7 Tg N a$^{-1}$.

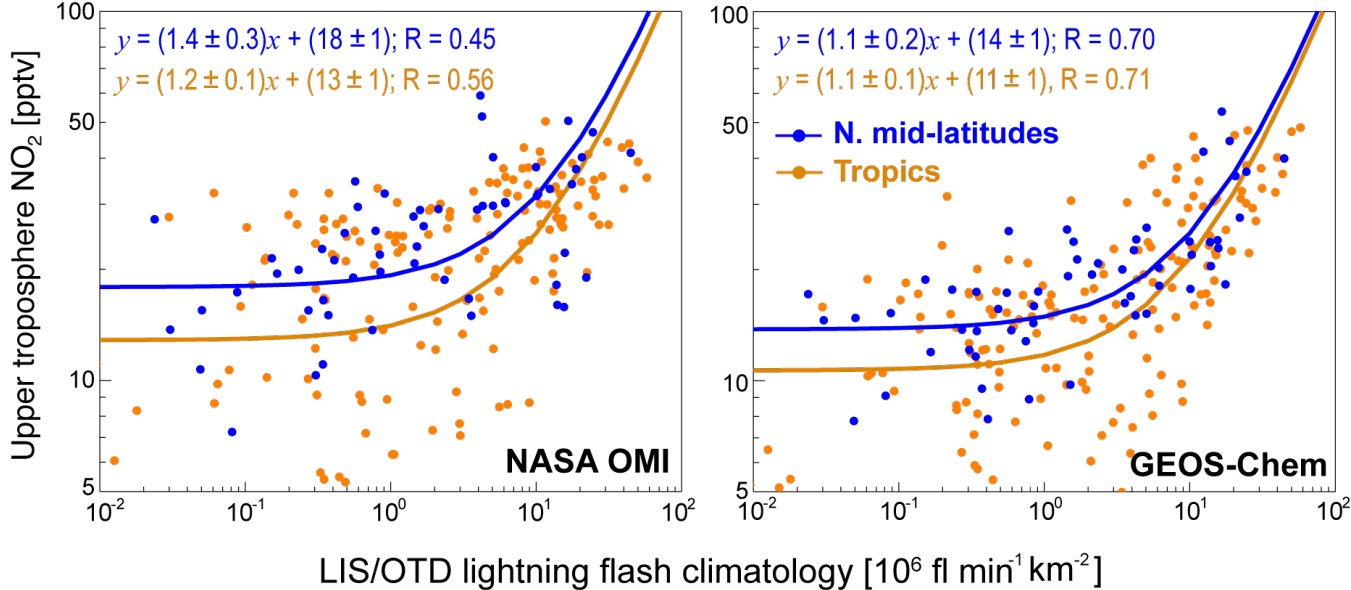

**Figure 6. Log-log relationship between upper troposphere NO₂ and lightning flash frequencies, and linear regression fits between the two. Individual points are coincident seasonal mean UT NO₂ from OMI (left) and GEOS-Chem (right) versus seasonal mean LIS/OTD lightning flash climatologies for coincident 20° × 32° gridsquares in the northern mid-latitudes (> 30°N; blue) and tropics (< 30°N; orange). Northern mid-latitude points exclude December-February that show poor correlation with lightning flashes (see text for details). Lines and legends show reduced major axis linear regression fits to the data with corresponding Pearson's correlation coefficients. The regression lines plot as curves on the log-log scale, highlighting the NO₂ background at low lightning flash rates and the correlation of NO₂ with lightning at high flash rates.**

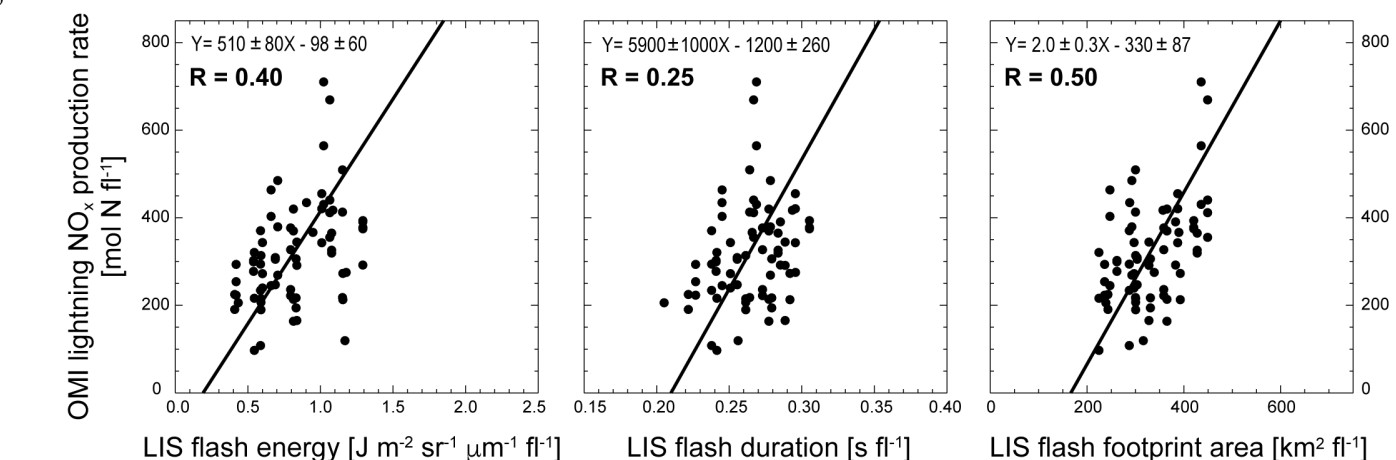

**Figure 7. Relationship between OMI and GEOS-Chem derived lightning NO$_x$ production rates and LIS lightning properties: energy (radiance), duration, and footprint area. Individual points are seasonal mean 20° × 32° gridsquares at 40°N-40°S.**

Properties of lightning flashes including energy, duration, and footprint area, have been retrieved from the OTD and LIS sensors (Beirle et al., 2014). The flash footprint area is the spatial extent of lightning detection events contributing to the flash (collection of local events) diagnosed by the satellite data. Figure 7 shows the relationship between OMI and GEOS-Chem derived lightning NO$_x$ production rates and LIS lightning properties from Beirle et al. (2014). The strongest correlation is with lightning extent (R = 0.50), followed by energy (R = 0.40). The correlation with flash duration is weak (R = 0.25). The relationships in Figure 7 suggest a dependence of lightning NO$_x$ production rates on lightning flash energy of $510 \pm 80$ mol N (J m$^{-2}$ sr$^{-1}$ μm$^{-1}$)$^{-1}$ and on flash footprint

area of $2.0 \pm 0.3$ mol N km$^{-2}$, possibly offering guidance for relating NO$_x$ yields to physical properties in global models rather than the current approach of assigning static values.

### 5. Conclusions

Measurements of NO$_x$ in the upper troposphere (UT) have mainly been from aircraft campaigns that are limited in space and time. Two new cloud-slicing UT NO$_2$ products from the Ozone Monitoring Instrument (OMI) produced by KNMI and NASA offer the potential to address uncertainties in our understanding of UT NO$_x$ sources. Here we intercompared these products, evaluated them with aircraft observations, and used them together with GEOS-Chem model simulations to demonstrate a dominance of lightning as a source of UT NO$_x$.


The KNMI and NASA UT NO$_2$ products use very different retrieval methods. Seasonal mean concentrations from the two products show weak global correlation at the $5° \times 8°$ (latitude $\times$ longitude) resolution of the NASA retrieval, with some improvement when the data are further averaged to $20° \times 32°$ (R = 0.5-0.7). At that resolution they show correlation with in situ aircraft observations of UT NO$_2$ over North America for different years (R = 0.56-0.64). The KNMI product is biased low by 38% relative to the aircraft

observations while the NASA product has no significant bias.

We find from the relationship of OMI UT NO$_2$ with LIS/OTD flash rates that most NO$_x$ in the upper troposphere is from lightning, except in the mid-latitudes in winter. The background NO$_2$ concentration in the absence of lightning is 10-20 pptv. The relationship suggests no difference in NO$_x$ yields per flash between the mid-latitudes and the tropics, in contrast to the higher yields at mid-

latitudes often assumed in models. We derive a global mean lightning NO$_x$ production rate per flash of $280 \pm 80$ mol N fl$^{-1}$, from which we infer a best estimate for the global lightning NO$_x$ emission of $5.9 \pm 1.7$ Tg N a$^{-1}$.

### Data Availability

Data from this work can be made available upon request: E. A. Marais for GEOS-Chem output, M. Belmonte-Rivas for KNMI OMI UT NO$_2$, S. Choi and J. Joiner for NASA OMI UT NO$_2$, and S. Beirle for LIS lightning properties.

### Competing Interests

The authors declare that they have no conflicts of interest.

### Author Contributions

EAM conducted model simulations, analysed and interpreted satellite, model, and aircraft data, and prepared the manuscript, DJJ provided supervisory guidance and assisted in the writing. SC, JJ, and MR-B retrieved the OMI UT NO$_2$ products, RCC aided in

interpreting aircraft observations., LTM contributed LIS/OTD lightning flash observations, SB contributed lightning flash properties, LS, VS, and LJ contributed updated GEOS-Chem code.

### Acknowledgements

This work was funded by the NASA Tropospheric Chemistry Program and a University of Birmingham Research Fellowship and NERC/EPSRC grant (EP/R513465/1) awarded to E. A. Marais. Model simulations were performed on the University of Birmingham's BlueBEAR High Performance Cluster. The authors would like to thank the BlueBEAR support team for IT and HPC support.

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
