# Peer review of "Nitrogen oxides in the global upper troposphere: interpreting cloudsliced NO2 observations from the OMI satellite instrument"

_Atmospheric Chemistry and Physics, 2018_

## Short Comment (SC1) · 22 Jun 2018

Well written paper about the cloud slicing method.

I have a few comments below:

1) L120: What's the definition of correlation between coincident gridsquare (R)? How did you calculate that?

2) L205: What's the ratio between intra-cloud (IC) and cloud-to-ground (CG) lighting? This will also affect UT NO2.

3) L208: You mentioned that the modeled lightning flashes are redistributed to match

[Figure]

LIS/OTD HRMC. How did you redistribute these flashes? Because lightning NOx can affect chemical reactions. Is this method online?

4) L210: It's better to explain the origin of both lightning production rates.

5) L215: How about the result of adjusting the production rate when compared with OMI?

6) L237: Did you exclude contamination of UT NO2 like southern Africa and northeast China when calculating the relationship?

---

## Referee Comment (RC1) · Anonymous Referee #1 · 27 Jul 2018

The manuscript by Marais et al. presents an application of satellite-derived NO2 data for the upper troposphere to diagnose NOx sources with the aid of the GEOS-Chem model. Two such products from the OMI instrument on the Aura satellite are evaluated in relation to aircraft data. The NASA product compared slightly better with the aircraft data than did the KNMI product. Therefore, the NASA product is used in comparison with GEOS-Chem output. This comparison suggests that the lightning NOx production in the model is too large, and the authors have scaled it down to better match the OMI-based data. The authors determined that there was no evidence to suggest larger NOx production efficiency per flash in the mid-latitudes than in the tropics. I have some concern about the level of detail of the lightning NOx emissions that are derived as I

describe below. I consider this a major revision. Otherwise, my comments are minor.

Neither of the OMI-derived UT NO2 products compared well with aircraft data. The better correlation (0.64) was with the NASA 450 - 280 hPa product at very coarse (20 x 32 degree resolution). However, this means Rˆ2 = 0.4 and that the satellite-based data only capture 40% of the variance seen in the aircraft data averaged to this resolution. Is this really good enough to constrain a global chemistry model? If one assumes there is sufficient meaning in these data, the comparison with GEOS-Chem suggests that the lightning NOx emission per flash in the mid-latitudes should be reduced from 500 to 260 moles/flash, leading to an overall lightning source strength reduction from 6.5 to 5.5 TgN/year. However, the authors go on to scale the lightning production per flash upward or downward for each 20 x 30 degree grid cell. Any discrepancy between the OMI UT data and GEOS-Chem is being attributed to differences in NOx production efficiency per flash. Given the relatively poor comparison between OMI and the aircraft data and uncertain model UT NOy chemistry, I think this is taking the analysis too far. It is a real stretch to quantitatively believe the values given in Figure 7 and in lines 259 - 269. I would suggest eliminating Figure 7 and perhaps just comparing the derived NOx production per flash values for mid-latitudes as a whole and tropics as a whole. Figure 8 could stay, as although it contains individual grid cell value of NOx production efficiency, it does not contain specific regional values that someone might quote.

Minor Comments:

Introduction section: The authors need to add some more background material on previous uses of OMI (and earlier satellites) data for diagnosing lightning NOx production. The prior literature is in 2 categories: satellite data and model analyses (Boersma et al., 2005; Martin et al., 2007) and satellite-alone analysis (Beirle et al., 2010; Bucsela et al., 2010; Pickering et al., 2016)

line 48: Add Allen et al., 2010 after Tost et al.

lines 52-53: The 100-500 mol N/flash and 3-7 TgN/yr do not match. If one assumes

the OTD/LIS climatological 46 flashes per second, 100 mol N/flash is about 2 TgN/yr and 500 mol N/flash is about 10 TgN/yr.

line 80: OMI was launched in July 2004.

line 90: For what pressure range was the Choi et al. (2014) product? Are there any other differences between that product and the NASA product used here?

lines 94-95: How well does TM4 do at these estimates?

line 101: I don't understand how this difference yields a column for 330-450 hPa.

line 158: ....lightning NOx emissions and convective transport of boundary layer pollution....

line 212: What percentage is this?

lines 216 - 222: Are these comparisons for the model with 6.5 TgN?

line 221: domain average UT NO2 is 19% lower than aircraft data. The opposite bias is present in comparing the model with OMI. Which should you believe?

line 242: Why would this be the case?

line 259: OMI-derived and GEOS-Chem lightning NOx production

line 289: 5.6 TgN/yr doesn't match the 5.5 TgN/year mentioned in line 214

---

## Referee Comment (RC2) · Anonymous Referee #2 · 15 Aug 2018

In this manuscript, upper tropospheric NO2 columns from two different cloud slicing approaches applied to one year of OMI data are used to study the impact of lightning on upper tropospheric NO2. First, data from the two retrievals is compared with each other and with aircraft observations taken over North America in the period March – August of the same year. Data from the NASA algorithm are then compared to GEOSchem model data for two seasons. Scatter plots of NO2 columns from both model and satellite retrievals against lighting flashes from the LIS/OTD climatology are compared and the conclusion is drawn, that UT NO2 from OMI is largely dominated by lightning NOx. Finally, spatially and seasonally resolved maps of NOx production per flash are computed from the ratio of retrieved to modelled UT NO2, and the dependency of these

production rates on LIS lightning properties is evaluated.

Measurements of NO2 in the UT are sparse and the use of satellite data for validation of model results in this important atmospheric region is of high scientific interest. The approach taken by the authors is interesting and the manuscript overall well written, although I would have hoped to get more details on what exactly was done in many places.

Nevertheless, the paper leaves me a bit helpless as my impression is, that combining the uncertainties of the individual steps taken in this analysis will make the results basically worthless. More specifically,

- the two retrievals which are based on the same data and on quite similar assumptions lead to very different results on UT NO2,
- the comparison with airborne measurements shows only broad agreement, and that only if data are averaged over large areas,
- the conclusion that the main driver for the observed UT NO2 variability is lightning Is probably correct in general but clearly not for individual points in Fig. 6,
- computing NOx emission rates per flash by taking ratios between model and measurement in the scattered distributions shown in Fig. 6 seems really optimistic to me.

I'm also surprised by the briefness of the discussion of the log-linear relationship found between lightning frequency and NO2. Is this a known fact, and is there an explanation for it? The fact that this relationship is not so clear in GEOS-chem data would not lead me to the conclusion that NOx lifetime is lower at high lightning frequencies (how would that follow from the lightning parametrisation used? Are non-linear effects really expected at the relatively coarse resolution of the model?) and that NO2 observations are uncertain at low concentrations (there are no observations in the model

**ACPD**
figure). I would rather suspect that other factors such as transport, vertical mixing, and chemistry are also important drivers of upper tropospheric NO2 in addition to lightning, which would explain that very large changes in lightning frequency are needed to see moderate changes in UT NO2.

The variations in NOx production per flash shown in Fig. 7 are large in many places, and would be important input for global modelling studies. However, an error bar is needed for these numbers before they can be used, and maybe this is the reason why the authors don't mention them in abstract and conclusions.

In summary, I cannot recommend this paper for publication in the current form. Before it can be published, the authors need to add more detail on the individual steps of the analysis and the data used, they need to provide uncertainty estimates and explain how they were derived, and also should add more discussion on how the fact that lightning is not the only factor affecting UT NO2 impacts on their results and conclusions.

**Minor comments**

- Introduction / beginning of section 2: There is a lot of repetition here, please read again and shorten where possible.
- page 3, line 97: Is aerosol really accounted for in the NO2 air mass factors, and if so, how?
- page 3, line109: What are near-Lambertian clouds?
- page 4, line 124: Why is the slant column offset affecting the UT NO2 data I thought this is cancelled by the stratospheric correction?
- page 4, line 133: How do the authors know which signatures in the figures are real, and which linked to misrepresentation of lower tropospheric signals?
- page 7, line 199: I understand that aircraft measurements are screened for stratospheric air masses in tropospheric applications. However, here data are compared to satellite retrievals, and these will - as far as I understand – include such stratospheric air masses if they are in the right pressure range above a cloud. I therefore wonder if this screening really makes sense here.
- page 7, Fig. 4. It is unfortunate, that here another time period is shown than in Fig. 1. As airborne data is collected over a period of 6 months, seasonal variability in UT NO2 could play a role in the comparison to satellite retrievals. I'd therefore suggest to show all 4 seasons in Fig. 1 or at least to add this figure to the Appendix / Supplement.
- page 8, Fig. 5. Again, I would suggest to add the other seasons as well.

---

## Author Comment (AC1) · 27 Sep 2018

**RESPONSES TO REVIEWERS AND X. ZHANG**

Ms. Ref. No.: Atmos. Chem. Phys. Discuss., doi:10.5194/acp-2018-556.

Title: Nitrogen oxides in the global upper troposphere: interpreting cloud-sliced NO2 observations from the OMI satellite instrument

Journal: Atmos. Chem. Phys. Discuss.

Reviewer comments are in blue. Responses are in black and include line numbers consistent with the updated manuscript with changes tracked. The manuscript starts on the 8th page of this pdf and the supplementary material requested by Reviewer #2 starts on the 25th page.

**Responses to Anonymous Referee #1:**

The manuscript by Marais et al. presents an application of satellite-derived NO2 data for the upper troposphere to diagnose NOx sources with the aid of the GEOS-Chem model. Two such products from the OMI instrument on the Aura satellite are evaluated in relation to aircraft data. The NASA product compared slightly better with the aircraft data than did the KNMI product. Therefore, the NASA product is used in comparison with GEOS-Chem output. This comparison suggests that the lightning NOx production in the model is too large, and the authors have scaled it down to better match the OMI-based data. The authors determined that there was no evidence to suggest larger NOx production efficiency per flash in the mid-latitudes than in the tropics. I have some concern about the level of detail of the lightning NOx emissions that are derived as I describe below. I consider this a major revision. Otherwise, my comments are minor.

**Major Comment:**

Neither of the OMI-derived UT NO2 products compared well with aircraft data. The better correlation (0.64) was with the NASA 450 - 280 hPa product at very coarse (20 x 32 degree resolution). However, this means R\*2 = 0.4 and that the satellite-based data only capture 40% of the variance seen in the aircraft data averaged to this resolution. Is this really good enough to constrain a global chemistry model? If one assumes there is sufficient meaning in these data, the comparison with GEOS-Chem suggests that the lightning NOx emission per flash in the mid-latitudes should be reduced from 500 to 260 moles/flash, leading to an overall lightning source strength reduction from 6.5 to 5.5 TgN/year. However, the authors go on to scale the lightning production per flash upward or downward for each 20 x 30 degree grid cell. Any discrepancy between the OMI UT data and GEOS-Chem is being attributed to differences in NOx production efficiency per flash. Given the relatively poor comparison between OMI and the aircraft data and uncertain model UT NOy chemistry, I think this is taking the analysis too far. It is a real stretch to quantitatively believe the values given in Figure 7 and in lines 259 - 269. I would suggest eliminating Figure 7 and perhaps just comparing the derived NOx production per flash values for mid-latitudes as a whole and tropics as a whole. Figure 8 could stay, as although it contains individual grid cell value of NOx production efficiency, it does not contain specific regional values that someone might quote.

Thank you for pointing this out. We have removed Figure 7 and now only discuss aggregated (tropics, midlatitudes, and global) lightning NOx production rates derived with OMI and GEOS-Chem (lines 359-364).

**Minor Comments:**

Introduction section: The authors need to add some more background material on previous uses of OMI (and earlier satellites) data for diagnosing lightning NOx production. The prior literature is in 2 categories: satellite data and model analyses (Boersma et al., 2005; Martin et al., 2007) and satellite-alone analysis (Beirle et al., 2010; Bucsela et al., 2010; Pickering et al., 2016)

Thank you for your suggestion. Studies listed in the comment include other  $NO_2$  sensors (GOME and SCIAMACHY) and so we provide a general background to the application of tropospheric column  $NO_2$ , rather than specific to OMI, to estimate lightning  $NO_x$  emissions and production rates (lines 69-72).

line 48: Add Allen et al., 2010 after Tost et al. Added (line 56).

lines 52-53: The 100-500 mol N/flash and 3-7 TgN/yr do not match. If one assumes the OTD/LIS climatological 46 flashes per second, 100 mol N/flash is about 2 TgN/yr and 500 mol N/flash is about 10 TgN/yr. Thank you for pointing out this contradiction. We now clarify that higher production rates

are typically applied to the northern midlatitudes than the tropics (line 61).

line 80: OMI was launched in July 2004.

This has been corrected (line 106).

**line 90: For what pressure range was the Choi et al. (2014) product? Are there any other differences between that product and the NASA product used here?**

We now specify the range of the mid-tropospheric product (900-650 hPa) (line 81) and also list the other differences between these products: cloud radiance fraction, number of OMI pixels, effective cloud scene pressure, and the cloud pressure product (lines 143-144, lines 159-160).

**lines 94-95: How well does TM4 do at these estimates?**

It is apparent in Figure 1 that TM4 is not able to correct for contamination over scenes with large surface sources and intermediate cloud fraction (southern Africa biomass burning in June-August). Rather than identify flaws in TM4 specifically, we reiterate that Belmonte-Rivas et al. (2015) caution that the correction relies on accurate simulation of NO2. (lines 177-178 and 180-181)

**line 101: I don't understand how this difference yields a column for 330-450 hPa.**

The assumption for both products is that the  $NO_2$  concentration is uniform across the pressure range of interest. For clarity, we now state the pressure centre and pressure range of each product (lines 127 and 132).

*line 158: ....lightning NOx emissions and convective transport of boundary layer pollution....* Added (line 216).

**line 212: What percentage is this?**

About 20-40% overestimate. This is now provided (line 277-278).

**lines 216 - 222: Are these comparisons for the model with 6.5 Tg N?**

We now clarify that GEOS-Chem simulations that follow use a single global lightning  $NO_x$  production rate of 260 mol N per flash (lines 278-279).

**line 221: domain average UT NO2 is 19% lower than aircraft data. The opposite bias is present in comparing the model with OMI. Which should you believe?**

We now clearly state that the model simulations from Figure 3 onwards use 260 mol N per flash everywhere (to address the previous comment).

**line 242: Why would this be the case?**

We reanalyzed the data to obtain a linear relationship between UT NO2 and lightning flashes to address one of the major comments by reviewer #2. This updated analysis is shown in Figure 6 in the manuscript and discuss the implications for UT NOx (lines 330-337).

*line 259: OMI-derived and GEOS-Chem lightning NOx production.* Changed throughout (lines 361, 485, Figure 7 caption).

line 289: 5.6 TgN/yr doesn't match the 5.5 TgN/year mentioned in line 214. Fixed (lines 26, 507).

**Responses to Anonymous Referee #2:**

In this manuscript, upper tropospheric NO2 columns from two different cloud slicing approaches applied to one year of OMI data are used to study the impact of lightning on upper tropospheric NO2. First, data from the two retrievals is compared with each other and with aircraft observations taken over North America in the period March –August of the same year. Data from the NASA algorithm are then compared to GEOS-Chem model data for two seasons. Scatter plots of NO2 columns from both model and satellite retrievals against lighting flashes from the LIS/OTD climatology are compared and the conclusion is drawn, that UT NO2 from OMI is largely dominated by lightning NOx. Finally, spatially and seasonally resolved maps of NOx production per flash are computed from the ratio of retrieved to modelled UT NO2, and the dependency of these production rates on LIS lightning properties is evaluated.

Measurements of NO2 in the UT are sparse and the use of satellite data for validation of model results in this important atmospheric region is of high scientific interest. The approach taken by the authors is interesting and the manuscript overall well written, although I would have hoped to get more details on what exactly was done in many places.

Nevertheless, the paper leaves me a bit helpless as my impression is, that combining the uncertainties of the individual steps taken in this analysis will make the results basically worthless. More specifically,

• the two retrievals which are based on the same data and on quite similar assumptions lead to very different results on UT NO2,

The two products follow very different retrieval steps. We now iterate that this is the case in Section 2 and point readers to retrieval details that follow (lines 115).

• the comparison with airborne measurements shows only broad agreement, and that only if data are averaged over large areas,

The intention in this work is to evaluate whether and at what temporal and spatial resolutions two new satellite-derived products provide useful information about global UT NO2, and which of the two retrieval approaches leads to data that is consistent with aircraft observations. This guides future retrievals with higher resolution instruments like TROPOMI. We now state this in Conclusions (lines 501-502).

• the conclusion that the main driver for the observed UT NO2 variability is lightning Is probably correct in general but clearly not for individual points in Fig. 6,

We now soften this claim in the Abstract (line 24).

• computing NOx emission rates per flash by taking ratios between model and measurement in the scattered distributions shown in Fig. 6 seems really optimistic to me.

We have removed Figure 7 (following the recommendation of Reviewer #1) and now only evaluate aggregated (midlatitudes, tropics, global) lightning NOx production rates.

I'm also surprised by the briefness of the discussion of the log-linear relationship found between lightning frequency and NO2. Is this a known fact, and is there an explanation for it? The fact that this relationship is not so clear in GEOS-Chem data would not lead me to the conclusion that NOx lifetime is lower at high lightning frequencies (how would that follow from the lightning parametrization used? Are non-linear effects really expected at the relatively coarse resolution of the model?) and that NO2 observations are uncertain at low concentrations (there are no observations in the model figure). I would rather suspect that other factors such as transport, vertical mixing, and chemistry are also important drivers of upper tropospheric NO2 in addition to lightning, which would explain that very large changes in lightning frequency are needed to see moderate changes in UT NO2.

Thank you for your comment. We now present the linear relationship between OMI UT NO2 and lightning flashes (Figure 6) and discuss the implication for UT NOx (lines 330-337).

The variations in NOx production per flash shown in Fig. 7 are large in many places, and would be important input for global modelling studies. However, an error bar is needed for these numbers before they can be used, and maybe this is the reason why the authors don't mention them in abstract and conclusions.

We have removed this figure (following the recommendation of Reviewer #1) and now only evaluate aggregated (midlatitudes, tropics, global) lightning NOx production rates.

In summary, I cannot recommend this paper for publication in the current form. Before it can be published, the authors need to add more detail on the individual steps of the analysis and the data used, they need to provide uncertainty estimates and explain how they were derived, and also should add more discussion on how the fact that lightning is not the only factor affecting UT NO2 impacts on their results and conclusions.

Thank you for your comment. We have responded to major comments above and minor comments below to address these concerns.

**Minor Comments:**

Introduction / beginning of section 2: There is a lot of repetition here, please read again and shorten where possible.

We have deleted repetitive statements (apparent in the tracked changes on lines 30, 72, 76, 82).

**page 3, line 97: Is aerosol really accounted for in the NO2 air mass factors, and if so, how?**

Yes. We now reference Boersma et al. (2004) (line 122) that is in turn referenced by Belmonte-Rivas et al. (2015) when describing the air mass factor calculation.

**page 3, line109: What are near-Lambertian clouds?.**

We now define these clouds with uniform scatter that are optically thick and geometrically thin (line 137).

**page 4, line 124: Why is the slant column offset affecting the UT NO2 data – I thought this is cancelled by the stratospheric correction?**

We have removed this statement to avoid confusion (line 170).

**page 4, line 133: How do the authors know which signatures in the figures are real, and which linked to misrepresentation of lower tropospheric signals?**

We are guided by the location of lightning flashes to determine which features are real in Figure 1 and by prior information about large surface sources of pollution to determine which features are associated with contamination.

page 7, line 199: I understand that aircraft measurements are screened for stratospheric air masses in tropospheric applications. However, here data are compared to satellite retrievals, and these will - as far as I understand – include such stratospheric air masses if they are in the right pressure range above a cloud. I therefore wonder if this screening really makes sense here.

Both the aircraft and satellite products exclude stratospheric contributions, and so the comparison is consistent. We now clarify that the OMI satellite product is tropospheric  $NO_2$  only (line 74). We already state that the KNMI product removes stratospheric  $NO_2$  by subtracting the stratospheric contribution in the retrieval (lines 121-122). We now elaborate that the stratospheric contribution in the NASA product is removed when differencing two nearby partial columns, as  $NO_2$  aloft is assumed uniform (lines 134-135 and lines 141-142).

page 7, Fig. 4. It is unfortunate, that here another time period is shown than in Fig. 1. As airborne data is collected over a period of 6 months, seasonal variability in UT NO2 could play a role in the comparison to

satellite retrievals. I'd therefore suggest to show all 4 seasons in Fig. 1 or at least to add this figure to the Appendix / Supplement.

Thank you for your suggestion. We now include the other seasons in supplemental Figure S1 (referenced in lines 163, 168, and 180).

page 8, Fig. 5. Again, I would suggest to add the other seasons as well. We now also include these in supplemental Figure S2 (referenced in lines 291-292).

**Responses to X. Zhang Interactive Comment:**

Well written paper about the cloud slicing method.

I have a few comments below:

1) L120: What's the definition of correlation between coincident gridsquare (R)? How did you calculate that?

We now specify in the text that this is the Pearson's correlation to determine spatial consistency between the two products in each season (lines 165-166).

2) L205: What's the ratio between intra-cloud (IC) and cloud-to-ground (CG) lighting? This will also affect UT NO2.

The GEOS-Chem parameterization does not distinguish  $NO_x$  production from between and within clouds and cloud-to-ground lightning (now stated in lines 279-281). Different flash types likely influence the amount of  $NO_x$  generated per flash (Schumann and Huntrieser, 2007), but models have poor predictive capability for the location of lightning flashes and by extension the type of flash (Murray, 2016).

3) L208: You mentioned that the modeled lightning flashes are redistributed to match LIS/OTD HRMC. How did you redistribute these flashes? Because lightning NOx can affect chemical reactions. Is this method online?

Scaling factors based on the discrepancy between the model estimate of lightning flashes and LIS/OTD monthly climatologies are used to correct for biases in the magnitude of the spatial distribution of lightning flashes (Murray et al., 2012). We have specified this for clarity (line 267).

4) L210: It's better to explain the origin of both lightning production rates.

We have rewritten this section for clarity (lines 275-281).

5) L215: How about the result of adjusting the production rate when compared with OMI?

This line doesn't correspond to any text, so we are unsure what is being asked.

6) L237: Did you exclude contamination of UT NO2 like southern Africa and northeast China when calculating the relationship?

We now explicitly state in the text that observations over northeast China are not considered in the comparison (lines 331-333). We only consider the NASA product in Figure 6 that has no observations over southern Africa in June-August (Figure 1).

**Deleted:** Here we use two new satellite products of UT NO2 mixing ratios from the Ozone Monitoring Instrument (OMI), together with in situ aircraft measurements and the GEOS-Chem chemical transport model, to assess current understanding of UT NOx sources.

... [1]

[revised manuscript text omitted]

Figure 1 compares seasonal mean UT NO2 from the two satellite products in December-February and June-August. KNMI NO2 is gridded to the NASA coarse grid. Data for March-May and September-November are in the Supplement (Figure S1), KNMI NO2 has greater coverage than the NASA product, due to a lower cloud fraction threshold in the retrieval. The two products exhibit very different spatial features. Spatial correlation between the two products (Pearson's correlation coefficient between coincident gridsquares) is R = 0.41 in December-February, R = 0.38 in June-August, There is marginal improvement in the correlation with further spatial averaging. At 20° × 32° we find R = 0.50 in December-February and R = 0.45 in June-August. The correlation only increases substantially in September-November from R = 0.49 at 5° × 8° (Figure S1) to R = 0.66 at 20° × 32°. KNMI is systematically lower than NASA in all seasons for coincident gridsquares, varying from 16% lower in June-

170 August to 48% lower in December-February at  $20^{\circ} \times 32^{\circ}$ .

160

165

180

KNMI OMI upper troposphere NO2 (2006)

Figure 1. Upper troposphere (UT) NO2 from the OMI satellite instrument. Seasonal mean UT NO2 from KNMI in 2006 at 330-450 hPa (top) is compared to NASA in 2005-2007 at 280-450 hPa (bottom). Data are at 5° × 8° horizontal resolution
 for December-February (left) and June-August (right). Grey areas indicate no data and, for NASA, scenes with fewer than 50 measurements.

Contamination of UT NO2 from below the cloud may still be present in the datasets despite attempts to correct for this using the TM4 model in the case of KNMI and by only considering very cloudy scenes in the case of NASA. These include a large enhancement in KNMI NO2 (> 90 pptv) over southern Africa in June-August when there is intense biomass burning, and the NO2 hotspot over northeast China in all seasons in both products (Figures 1, S1). Belmonte-Rivas et al. (2015) caution that the contamination correction in the KNMI product relies on accurate simulation of NO2 vertical distribution.

| Deleted: re                              |
|------------------------------------------|
| Deleted: and NASA data are for 2005-2007 |
|                                          |
| Deleted: C                               |
| Deleted: coincident gridsquares is weak  |
| Deleted: )                               |
|                                          |

**3. Evaluation of OMI upper troposphere NO2 with aircraft observations**

- 195 The aircraft observations we use to evaluate the OMI NO2 products are from thermal-dissociation laser-induced fluorescence (TD-LIF) instruments (Day et al., 2002) for NASA DC8 aircraft campaigns over North America and Greenland in springsummer when there is a high density of measurement campaigns. These include INTEX-A, INTEX-B, ARCTAS, DC3, and SEAC4RS. Only INTEX-B is in the same year as the OMI products but we consider interannual variability to be only a small source of error. Measurements of NO2 from TD-LIF are susceptible to interference from decomposition of thermally unstable
- 200 reservoir compounds methyl peroxy nitrate (CH3O2NO2) and HNO4, in particular in the UT, where NO2 concentrations are relatively low, temperature gradients between the instrument inlet and ambient air are large, and reservoir compounds are abundant (Browne et al., 2011). Publicly available DC3 and SEAC4RS TD-LIF NO2 are already corrected for this interference. We apply a correction for the other campaigns using the relationship between temperature and percentage interference from Browne et al. (2011). Observed mean ambient air temperature in the UT during INTEX-A is 246 K, corresponding to 20% 205 interference. That for INTEX-B is 241 K (30% interference) and 236 K for ARCTAS (38% interference).

210

There are also NO2 observations from the recent NASA ATom campaign, and from the In-service Aircraft for a Global Observing System (IAGOS) commercial aircraft campaign (Berkes et al., 2017). These use chemiluminescence instruments that are also susceptible to interference. Chemiluminescence and TD-LIF NO2 are consistent during the SEAC4RS campaign for the altitude range considered in this work (6-9 km) (Travis et al., 2016), but the interference from chemiluminescence is challenging

to quantify, due to dependence also on the operator and instrument design that varies across campaigns (Reed et al., 2016).

Figure 2 shows the sampling extent of TD-LIF UT NO2 over North America and Greenland in spring-summer at 450-280 hPa. around the satellite overpass (11h00-16h00 LT) for scenes not influenced by the stratosphere (diagnosed with collocated 215  $ozone/CO > 1.25 mol mol^{-1}$  (Hudman et al., 2007)). Concentrations of UT NO2 exceed 80 pptv over the eastern US due to

lightning NOx emissions and convective transport of boundary layer pollution, and are

---

## Author Response (AR2)

**FOLLOW-UP RESPONSES TO REVIEWER #2**

Ms. Ref. No.: Atmos. Chem. Phys. Discuss., doi:10.5194/acp-2018-556.

Title: Nitrogen oxides in the global upper troposphere: interpreting cloud-sliced NO₂ observations from the OMI satellite instrument

Journal: Atmos. Chem. Phys. Discuss.

Reviewer comments are in blue. Responses are in black and include line numbers consistent with the updated manuscript with changes tracked (response to reviewer #2 and minor editorial updates). The manuscript starts on the 4ᵗʰ page of this pdf.

**Responses to Anonymous Referee #1:**

We would like to thank reviewer #2 for their second meticulous review of the paper. Their comments and careful thought of the paper are incredibly helpful and will no doubt improve the quality of the publication.

*1. line 60: Bucsela et al. (2010) should appear on the previous line.*

Updated.

*2. lines 99 -100: I still do not understand how taking the difference between the column covering 380 hPa and the tropopause and that covering 380 to 500 hPa yields the column centered at 380 hPa (330 - 450 hPa).*

We now clarify that UT NO₂ is obtained as the difference between two neighboring partial tropospheric columns retrieved above clouds that are at a mean pressure range of 330-450 hPa and 380-500 hPa (lines 99-100).

*3. line 107: The criterion of cloud radiance fraction > 0.7 allows some amount of signal from the lower troposphere to enter the cloud-sliced data set. Both Choi et al. (2014) and Pickering et al. (2016) favored CRF > 0.9 for this reason. Pickering et al. (2016) report a 5% high bias when using CRF >0.7 compared with CRF > 0.9. A paragraph is needed discussing all of the uncertainties involved in the OMI cloud-sliced NO2 product including this possible bias.*

Thank you for highlighting the sensitivity test by Pickering et al. (2016). Their results clearly demonstrate that the effect of different CRF thresholds of 0.9 and 0.7 is small (<5%) and not significant (derived NO$_x$ yields have large variability: $117 \pm 94$ mol fl$^{-1}$ using a CRF>70% and $112 \pm 67$ mol fl$^{-1}$). We now state this in the paper (lines 117 and 128-127).

*4. lines 257 - 262: The production rate derived here is very model dependent. The authors need to state this fact and that the values derived here are influenced by any error that exists in the UT NOy chemistry that may be in GEOS-Chem. Such possible errors have recently been discussed by Travis et al. (2016) and Silvern et al. (2018).*

We now state that the modelled and observed NO₂ are consistent over the pressure range of the OMI UT NO₂ product (6-9 km). The comparison between modelled and observed NO₂ by Silvern et al. (2018) is adapted in Figure 1 below and shows that the model reproduces the observed NO₂ across the altitude range of the OMI UT NO₂ product (lines 225-227). The bias reported by Travis and Silvern is for higher altitudes.

[Figure]

**Figure 1.** Comparison of TD-LIF and GEOS-Chem $NO_2$ during SEAC$^4$RS (adapted from Silvern et al., 2018). Two versions of GEOS-Chem are shown that test sensitivity of $NO_2$ vertical distribution to uncertainties in reaction kinetics and photolysis. Blue arrow shows the altitude range of the NASA OMI UT $NO_2$ product.

*5. page 3, line 97: Is aerosol really accounted for in the NO2 air mass factors, and if so, how?*

*Yes. We now reference Boersma et al. (2004) (line 122) that is in turn referenced by Belmonte-Rivas et al. (2015) when describing the air mass factor calculation.*

*I disagree. Please have another look at Boersma et al., 2004 last sentence of section 3.2. Aerosols are not explicitly taken into account in the DOMINO NO2 data product, only implicitly in partially cloudy scenes through the cloud correction applied. However, this correction has no effect on the cloudy scenes discussed here.*

Thank you for pointing us to the appropriate location in Boersma et al. (2004) that makes clear that aerosols are not directly accounted for in the DOMINO $NO_2$ AMF calculation. We have updated the text to reflect this (line 95).

*6. page 7, line 199: I understand that aircraft measurements are screened for stratospheric air masses in tropospheric applications. However, here data are compared to satellite retrievals, and these will -- as far as I understand – include such stratospheric air masses if they are in the right pressure range above a cloud. I therefore wonder if this screening really makes sense here.*

*Both the aircraft and satellite products exclude stratospheric contributions, and so the comparison is consistent. We now clarify that the OMI satellite product is tropospheric NO2 only (line 74). We already state that the KNMI product removes stratospheric NO2 by subtracting the stratospheric contribution in the retrieval (lines 121-122). We now elaborate that the stratospheric contribution in the NASA product is removed when differencing two nearby partial columns, as NO2 aloft is assumed uniform (lines 134-135 and lines 141-142).*

*This is not the point. The NASA product takes the difference in NO2 columns over neighbouring clouds having different cloud top. Any NO2 in between these altitudes will contribute to the retrieved UT mixing ratio, regardless of its origin. If stratospheric NO2 is brought down to the UT, the cloud slicing method will detect it as UT NO2 (as it should).*

The retrieval approach of Choi et al. (2014) only considers scenes with well-mixed $NO_2$, diagnosed with a vertical gradient threshold of 0.33 pptv hPa$^{-1}$ from the GMI chemical transport model. We also only consider scenes with more than 50 measurements at coarse resolution ($5° \times 8°$), so downwelling from the stratosphere that concerns the reviewer would have to persist throughout the gridsquare and over the whole season for multiple years (2005-2007) to bias the OMI UT $NO_2$. We also now state that the contribution of stratospheric injection, as estimated by Murray et al. (2012) using GEOS-Chem, is small (lines 240-242).

*7. Please add the uncertainty estimates for the NOx yield and global lightning NOx source given in the text also to the abstract and conclusions.*

The error in the $NO_x$ yield and global lightning $NO_x$ source is now stated in the Abstract and Conclusions. We have also included the estimate of global lightning $NO_x$ with the size of the uncertainty inferred from the relative size of the uncertainty in the $NO_x$ production rate (lines 286-289).

**References:**

Murray et al. (2012), doi:10.1029/2012jd017934.

[revised manuscript text omitted]